# Implementation of an Environmental Cleaning Protocol in Hospital Critical Areas Using a UV-C Disinfection Robot

**DOI:** 10.3390/ijerph20054284

**Published:** 2023-02-28

**Authors:** Beatrice Casini, Benedetta Tuvo, Michela Scarpaci, Michele Totaro, Federica Badalucco, Silvia Briani, Grazia Luchini, Anna Laura Costa, Angelo Baggiani

**Affiliations:** 1Department of Translational Research and the New Technologies in Medicine and Surgery, University of Pisa, 56126 Pisa, Italy; 2Hospital Management, University Hospital of Pisa, 56126 Pisa, Italy

**Keywords:** hospital environmental cleaning and disinfection, ultraviolet C light-emitting device, high-touch surfaces

## Abstract

Improving the cleaning and disinfection of high-touch surfaces is one of the core components of reducing healthcare-associated infections. The effectiveness of an enhanced protocol applying UV-C irradiation for terminal room disinfection between two successive patients was evaluated. Twenty high-touch surfaces in different critical areas were sampled according to ISO 14698-1, both immediately pre- and post-cleaning and disinfection standard operating protocol (SOP) and after UV-C disinfection (160 sampling sites in each condition, 480 in total). Dosimeters were applied at the sites to assess the dose emitted. A total of 64.3% (103/160) of the sampling sites tested after SOP were positive, whereas only 17.5% (28/160) were positive after UV-C. According to the national hygienic standards for health-care setting, 9.3% (15/160) resulted in being non-compliant after SOP and only 1.2% (2/160) were non-compliant after UV-C disinfection. Operation theaters was the setting that resulted in being less compliant with the standard limit (≤15 colony-forming unit/24 cm^2^) after SOP (12%, 14/120 sampling sites) and where the UV-C treatment showed the highest effectiveness (1.6%, 2/120). The addition of UV-C disinfection to the standard cleaning and disinfection procedure had effective results in reducing hygiene failures.

## 1. Introduction

The role of the environment, particularly of high-touch surfaces in the patient’s room (e.g., bedrails, over-bed tables, and call-buttons) and reusable care equipment that is moved between rooms, has been demonstrated to be fundamental in the transmission of healthcare-related infections. Together with standard precautions and the application of good practices in invasive procedures, environmental cleaning and disinfection represents one of the three pillars of infection risk prevention in healthcare settings.

Environmental contamination in healthcare settings can play a key role in the transmission of HAIs [1,2]. It was estimated that 20% of healthcare-associated infections originate from contaminated environmental surfaces [3].

Several studies have shown that the manual cleaning and disinfection of surfaces in hospitals is suboptimal [4,5,6] and that 50% or more hospital surfaces may go untouched and uncleaned during terminal room disinfection. In addition, it was estimated that approximately 5–30% remain contaminated, despite the application of adequate cleaning and disinfection protocols [7]. The ineffectiveness of existing cleaning and disinfectant formulations can be attributed to the presence of dry biofilm, where the survival of vegetative bacteria for long periods has been demonstrated [8,9,10].

Therefore, the enhancement of cleaning and disinfection protocols is strongly recommended for high-touch surfaces, particularly in settings at risk of outbreaks transmission of multi-drug resistance microorganisms (MDROs) [11].

Environmental cleaning considered not as a standalone bundle but as part of a multimodal strategy is one of the eight core components of an effective infection prevention and control (IPC) strategy [12].

Hospital environmental hygiene is a complex process because it is influenced by several variables, such as the type of surface, incorrect disinfectant contact times, excessive dilution of disinfectant solutions, and potential biocide/antibiotic cross-resistance. In addition, it has been shown that the use of contaminated cloths and/or solutions promotes the spread of microorganisms between different environments [2,4,7,13].

The cleaning procedure is not only dependent on the chemicals used but also on the personnel performing it. As reported by Toffollutti et al., there are differences between housekeeping and outsourcing cleaning staff in ensuring adequate levels of hygiene; outsourcing cleaning services was associated with a greater incidence of meticillin-resistant Staphylococcus aureus (MRSA) and worse patient perceptions of cleanliness [14]. On the other hand, the high turnover of cleaning staff can lead to poorly motivated or less trained people [15,16,17]. In a recent study of Daniels, the importance of cleaning-staff-specific training with specialty certifications, also regarding infection prevention, as well as the identification of responsibilities and the recognition of the role played by cleaning staff in a healthcare setting, was highlighted [18].

In contexts where this is difficult to achieve, it is essential to enhance terminal disinfection, possibly by introducing optimized automated no-touch disinfection methods such as UV-C treatment [19,20]. This technology provides short disinfection times (the disinfection cycle takes approximately 5 min), is easy to use, has minimal need for the special training of staff, and, unlike hydrogen peroxide vapor systems, ventilation and air conditioning do not need to be disabled and the rooms do not need to be sealed. This technology also has disadvantages. Single rooms in hospital are rare and the treatment is not appliable in multi-bedrooms as, to guarantee people’s safety, it has to operate in the absence of an operator/patient. In addition, the material compatibility has to be checked and it has to be applicated after cleaning because organic material could interfere with the effectiveness of the disinfection treatment [21].

The effectiveness against MDRO and reduction in the bioburden from surfaces have been reported by several studies that have assessed the effectiveness of UV devices in decontaminating hospital rooms following the discharge of a patient colonized or infected with an MDRO [5,22,23,24,25,26].

The recent measures of the containment and management of the COVID-19 pandemic have further underlined the importance of environmental cleaning and disinfection.

Several factors influence the risk of environmental contamination (technological, logistical, construction), among which are primarily the procedures of cleaning and disinfection, such as methods, frequencies, and used products [27].

Among the appliable strategies for the improvement of cleaning and disinfection practices are the use of new materials and/or disinfectants, the training and audit of operators, and the use of new automated technologies, which are becoming increasingly important. In particular, no-touch disinfection technologies have the great advantage of not being dependent on the operator, ensuring process repeatability. Furthermore, their effectiveness has been demonstrated even on sites that are difficult to reach with manual intervention. Their use complements but does not replace ordinary cleaning and disinfection protocols. In the past few years, ultraviolet disinfection systems have been widely investigated and used as a way to improve standard cleaning protocols. Currently, ultraviolet devices are automated in order to guarantee process repeatability and reduce human errors. The application of UV devices as an addition to traditional environmental cleaning has become increasingly common due to their effectiveness in reducing the environmental microbial burden in a shorter time compared to other technologies using chemical products [28]. COVID-19 pandemic UVC disinfection has been applied to several different settings, starting with COVID-19 patients’ room decontamination, or operating rooms [29,30]. The efficacy of this method was also widely demonstrated for the treatment of electromedical devices, such as computed axial tomography [31].

Over the past 80 years, several authors have demonstrated a significant reduction in the transmission of pathogens responsible for airborne diseases such as measles, influenza, and tuberculosis following UV-C air disinfection [32,33]. For this reason, during the COVID-19 pandemic, UV-C radiation was proposed to maximize the protection of the population against the airborne spread of SARS-CoV-2.

Given evidence of the stability of SARS-CoV-2 aerosols with high levels of infectivity for at least 3 h, effective air disinfection technologies could play an important role in reducing air viral contamination, allowing for a safer access to indoor spaces [34].

In that regard, the World Health Organization has reiterated the importance of using this type of disinfection after the adoption of the standard protocol for sanitizing hospital rooms, but, during the COVID-19 pandemic, the use of UVC radiation was also proposed immediately after the patient has been discharged in order to reduce the risk of the cleaning staff being infected, who can then safely apply the standard protocol [35].

The disinfection of the air with UV-C is performed by irradiating the upper-room air only, the whole room when unoccupied, or the air flowing inside the air-handling units. The study conducted by McGinn et al. demonstrated the feasibility of using a UV-C robotic system to disinfect both air and surfaces in a radiology environment, where it was two and four times faster than currently used chemical approaches [36].

Although more high-quality studies are needed, a good relation was observed between interventions used to improve the hospital environmental hygiene, such as UV-C disinfection, and a reduction in patient colonization or healthcare-associated infections [37].

The aim of this study was to evaluate the effectiveness of the Hyper Light UV-C Disinfection Robot (Mediland Enterprise Corporation) in reducing environmental microbial contamination when applied in addition to standard cleaning and disinfection protocols in critical areas with a high risk of transmission of healthcare infections.

## 2. Methods

A prospective open-labelled cross-over study was conducted in a 1158-bed teaching hospital in Italy from 14 April to 16 June 2021. To evaluate the effectiveness of the UV-C disinfection robot in reducing environmental contamination, sampling was performed in four different critical areas: 1 single occupancy ward room, 1 intensive care unit (ICU) isolation room occupied for a minimum of 48 h, and 2 operating theaters (OTs).

### 2.1. UV-C Disinfection Device

The UV-C disinfection robot (Mediland Enterprise Corporation, Taoyuan City, Taiwan) uses amalgam lamps (UV lamp NNI 300/147 XL Niederdruck VUV Strahler) and protective reflector technology to generate high-energy, broad-spectrum ultraviolet light (UV-C 100–280 nm). The manufacturer of the lamps declares in the technical data sheet that the lamps have a filter that blocks radiation between 180–185 nm, eliminating the possibility of producing ozone as a by-product of UV-C radiation. The UV-C device uses 5 min disinfection cycles and multiple positions with minimal distances from high-touch surfaces. Due to the use of high-intensity UV-C radiation, the device must operate in unoccupied rooms. There are multi-motion sensors that shut off the device if any movement is detected inside the room being disinfected or if the door is accidentally opened. When the robot operates in accordance with these procedures, the manufacturer declares that the amalgam lamps produce no ozone gas and leave no toxic residues.

### 2.2. Environmental Sampling Method

In each setting, 20 high-touch surfaces were selected. Baseline microbiologic samples were collected after patient discharge or after surgical activity and immediately after cleaning and disinfection standard operating protocol (SOP) or after UV-C treatment (Before SOP, After SOP, After SOP+UV-C).

Each room (ward and ICU room) was sampled once, whereas surgical theaters were sampled more times; globally, 160 sample series (Before SOP, After-SOP, After SOP+UV-C) were collected for a total of 480 samples. Of the 160 series, 20 were taken in the ward room, 20 in the ICU room (total 120 samples), and 120 in the surgical theaters (total 360 samples).

In this hospital, cleaning services were outsourced. According to the contract and the cleaning and disinfection standard operating protocol (SOP), during terminal disinfection, the housekeeping staff applied a chlorine-based detergent, Antisapril Detergent 10%, Angelini, followed by a chlorine-based disinfectant (Deornet Clor (ÈCOSÌ, Forlì-Cesena, Italy), active chlorine 2600 mgr/L) on furniture surfaces and electromedical devices.

After SOP, the UV-C robot was employed with multiple 5 min cycles: in the ward room, three for each bedside and one in the bathroom, three in the operating theaters (OTs) for each surgical table side, and two in the ICU rooms for each bedside. The two treatments were carried out in series and the sampling was carried out immediately after SOP and after SOP+UV-C.

According to ISO 14698-1, 55 mm diameter Rodac plates containing plate count agar (PCA) with neutralizers (VWR International PBI, Radnor, PA, USA) were used for the total viable count (TVC) enumeration. Contact plates were incubated aerobically at 37 °C for 48 h.

In the OT and ICU, the total microbial load and presence of pathogens were evaluated according to the hygienic standards proposed by the Italian Workers Compensation Authority, which recommended a limit of ≤50 CFU/24 cm^2^ for ICUs and ≤15 CFU/24 cm^2^ for operating theaters [38], whereas, in the ward rooms, the evaluation was conducted according to the standard proposed by Dancer et al. (≤125 CFU/24 cm^2^) [39].

On each sampled site, the emitted dose was evaluated by a dosimeter that is able to measure UV doses from 0 mJ/cm^2^ to 250 mJ/cm^2^ (FastCheck Mediland, Taoyuan City 33382, Taiwan), changing color from yellow to green and allowing for a quick visualization of disinfection performance.

### 2.3. Statistical Analysis

A pair-matched analysis was performed to evaluate the trend of each TVC compared to the hygienic limit for each type of setting after SOP and UV-C disinfection. The sign test was used (non-parametric) with two-sided significance levels *p* < 0.016 (fixed to 3 variables). The *p*-value to take into account is the one estimated by the two-sided test.

## 3. Results

We collected a total of 480 samples on high-touch surfaces: 160 after healthcare activity, 160 after the SOP, and 160 after the application of the UV-C treatment. There were 103/160 (64.3%) positive samples after SOP, whereas there were 28/160 (17.5%) positive samples after the treatment with UV-C.

In the OT, 58% (70/120) of the samples showed a TVC equal to 12.53 ± 28.33 CFU/cm^2^ after SOP, whereas only 20% (4/20) showed a TVC of 0.84 ± 3.18 CFU/cm^2^ after the application of UV-C treatment.

In the ICU isolation room, 80% (17/20) of the samples resulted in being positive after SOP, showing a TVC equal to 12.38 ± 28.30 CFU/cm^2^, but only 20% (4/20) were positive after UV-C treatment, with a TVC of 0.54 ± 1.74 CFU/cm^2^.

In the ward rooms, 85% of the samples resulted in being positive (17/20) after SOP, with a TV-C of 12.94 ± 30.44 CFU/cm^2^, but no samples were positive after the application of the UV-C disinfection robot.

The most contaminated surfaces, before and after SOP, were the patient’s bed, infusion pump, and tray table. On these last two surfaces, the mean microbial load after SOP increased compared to before, with values equal to 5 ± 0 CFU/cm^2^ before cleaning and 35 ± 49.4 CFU/cm^2^ after SOP; whereas, on the tray table, the mean load before cleaning was 17.3 ± 17.6 CFU/cm^2^, and 132.3 ± 82.5 CFU/cm^2^ after SOP. On the patient’s bed, the mean load was equal to 73.5 ± 136.2 CFU/cm^2^ and 44.2 ± 23.7 CFU/cm^2^.

In the OT and ICU, where the limit for hygiene quality is defined by national guidelines, respectively, 12% (14/120) and 5% (1/20) of the surface samples resulted in being non-compliant after SOP. In the ward rooms, according to the standard reported in the literature, 15% (3/20) of the sampled sites resulted in being non-compliant. All of the samples were compliant after UV-C disinfection. In all of the settings, 95% (152/160) of the sampled points received a medium UV-C dose of 200 mJ/cm^2^ and only 5% (8/160) received a dose of 50mJ/cm^2^. Of the latter, 6 were in the OT (6/120), 1 in the ICU (1/20), and 1 in the ward room (1/20).

Of particular interest was the observation that the TVC did not undergo significant variation after the application of SOP; rather, we observed an increase in TVC in 25 out of 120 samples collected in the OT, for which, the microbial load increased from 9.68 ± 18.82 CFU/cm^2^ before SOP to 10.68 ± 9.82 CFU/cm^2^ after SOP. The same anomaly was also found when analyzing the samples collected from the ICU isolation rooms and from the ward rooms after SOP: 20% (4/20) and 75% (15/20) of the sampled surfaces, respectively, showed an increase in the microbial amount from 4.5 ± 4.5 and 9.53 ± 9.37 CFU/cm^2^ before SOP to 12 ± 6.12 and 75.8 ± 56.71 CFU/cm^2^. These results were subsequently investigated through an audit that involved checking the cleaning and disinfection protocol used by the outsourced company, verifying how the company managed the washing of reusable cloths, and the conservation and dilution of the chemical products used. During the audit, some critical points were identified in the process that required corrective actions: although color codes were given for the cloths used in different settings, washing was carried out simultaneously in the same washing machine and, moreover, chemicals were left unsealed.

In this context, the addition of the UV-C use to the disinfection process certainly improved the hygiene level, but also highlighted critical issues in the SOP that needed to be corrected. The reduction in TVC after the UV-C treatment was statistically significant (*p* < 0.0005) (Figure 1) as shown by the quadratic regression analysis. (Figure 2).

## 4. Discussion

The results of our study are in close agreement with the previous research, which found a significant decrease in the bioburden in all investigated hospital settings, below the hygienic standard limits recommended. The dose emitted by the device has always been high, and higher than the values reported in the literature as effective against pathogens [40].

Although the dose measurement system used in our study (FastCheck Mediland, Taoyuan City 33382, Taiwan) is semi-quantitative, it allowed us to detect if the dose reached adequate values for disinfection and to repeat the application when the value was not satisfactory. However, each dose value exceeded the expected value. This demonstrates that the device was positioned correctly in order to avoid shadow areas, although it was necessary to properly place furniture and equipment to improve the irradiation of all surfaces. In the study conducted by Wong et al., the R-D Rapid Disinfector system (Steriliz, Rochester, NY, USA) was used in a tertiary care hospital to evaluate the incremental benefit of UV-C disinfection in isolation rooms after the discharge of infected patients. The employed robot used four detached sensors to directly measure the UV-C light [26].

During our study, microbial contamination beyond the limits set for selected hospital settings—in particular, in operating theaters, which require more restrictive hygiene standards—was detected (14/120, 12% of the sampled sites). After UV-C disinfection, all samples were negative, highlighting the good efficacy of the UV-C-implemented disinfection.

Our study highlighted some critical issues in the procedure adopted by the outsourcing cleaning service regarding the reprocessing of the reusable cloths and the management of chemicals with recorded bioburden values on surfaces that were even higher than those detected before cleaning.

Outsourced hospital cleaning services are often performed by low-paid staff that are poorly trained on infection control issues and have a low work motivation. All of this is reflected in the lower-quality level of cleaning compared to that carried out by hospital staff.

Despite several studies [6,14] highlighting this criticality, few mitigation strategies have been chosen to date. While it is known that hospital personnel have a greater perception of the significance of housekeeping as an important tool for reducing environmental microbial contamination, further efforts are needed to improve the training of personnel involved in environmental cleaning procedures. In Italy, there is no certification of the skills that these personnel must possess in order to be able to carry out these activities.

As reported by the WHO, there is no single best method for cleaning and disinfecting, but there are critical components that help to create and sustain a clean, safe environment that supports the safety of patients and healthcare professionals [12]. Among these, there is also environmental monitoring, which allows us to understand the current state of the cleanliness of the structure, identifying areas for improvement, including the choice of suitable cleaning and disinfection systems. 

Monitoring data feedback to cleaning staff through periodic audits can be a valid aid for improving cleaning and disinfection procedures and adherence to protocols. This strategy was adopted in our study and, thanks to the collaboration between the infection control team and the outsourced cleaning company, it was possible to identify the critical issues and take corrective actions. To this end, environmental monitoring is essential for process verification.

Each hospital should have an infection control team aiming to evaluate the risk factors involved in healthcare infection occurrences with a multidisciplinary and dynamic approach. Epidemiological infection control in hospital may detect all of the critical points of the healthcare procedures performed by nurses, healthcare workers, physicians, students, and external staff. This evaluation may also include the sanitization process and its management. An appropriate evaluation of the whole sanitization process, including the reprocessing of cleaning materials, would be the best practice.

In previous studies, UV-C disinfection has been employed as one of the infection control interventions in response to high rates of HAI or clusters of infections with a specific pathogen [41]. In our study, the use of UV-C disinfection was introduced in critical hospital areas following the detection of non-compliance with the standard cleaning and disinfection protocol, which was found to be the cause of environmental contamination and hygienic failures. Operating theaters are easier to treat with UV-C because disinfection can be performed before and after surgery. Single-bed hospital and ICUs rooms have the same versatility, whereas UV-C radiation is not suitable for use in promiscuous environments, where the proximity of other bed stations reduces use possibilities due to safety issues. Given the importance of ensuring the safety of environments hosting critical patients, such as ICUs, it would be worth evaluating strategies aiming to protect the hosted patients during UV-C irradiation.

Hospitalized patients’ security in intensive care units should be ensured using protective devices, which may be evaluated for human security during UV-C emission at specific dosages and times. Considering the lack of literature data about these possible strategies, the use of a UV-C robot in the presence of persons still represents a critical issue that may be taken into account for the development of safe and effective technologies.

Our study has some limitations. Firstly, the non-homogeneous sampling for all of the settings, where a greater number of analyses were dedicated to the OTs rather than the ICUs or patient rooms: it was not always possible to proceed with the UV-C treatment at the patient discharge because trained personnel were limited and not always available. This problem underlines the importance of having dedicated personnel identifiable as that of the outsourced cleaning company. The use of new technologies could in fact be included in outsourcing contracts as an improvement action in the event of non-compliance in the cleaning service. These personnel could be adequately trained not only on the cleaning protocols to be adopted but also on infection control issues.

In our study, we evaluated the efficacy of the UV-C treatment only on the reduction in mesophilic growth that mainly represents human contamination, but not on specific pathogens of interest; however, the reduction was very significant, suggesting that this system can easily eliminate pathogens as well. We have previously demonstrated that the adoption of an automated UV-C-disinfection robot in the enhancement of SOP in high-risk settings was successful in reducing pathogens on high-touch surfaces, improving the patient’s safety [7].

Moreover, it would be important to evaluate the advantages of implemented cleaning and disinfection protocols with the use of UV-C devices in areas hosting fragile and vulnerable patients, e.g., in intensive care units. The principal limitation is the continuous presence of people in this type of setting. For this reason, it would be worth evaluating strategies aimed at protecting the patient’s safety using protective devices, which may be evaluated for human security during UV-C emission at specific dosages and times.

## 5. Conclusions

In settings where hospital cleaning services are assigned to private sector contractors, the use of new no-touch technologies could be considered in outsourcing contracts as an improvement action in the event of non-compliance in the cleaning service. The dedicated personnel should be adequately trained not only on the cleaning protocols to be adopted but also on infection control issues, underlining the key role of cleaning procedures.

UV-C was shown to be a valid disinfection method because of the effectiveness shown after a short exposure time. Moreover, the UV-C robot can automatically record the disinfection parameter, permitting the monitoring of the process quality given the traceability of the procedures.

## Figures and Tables

**Figure 1 ijerph-20-04284-f001:**
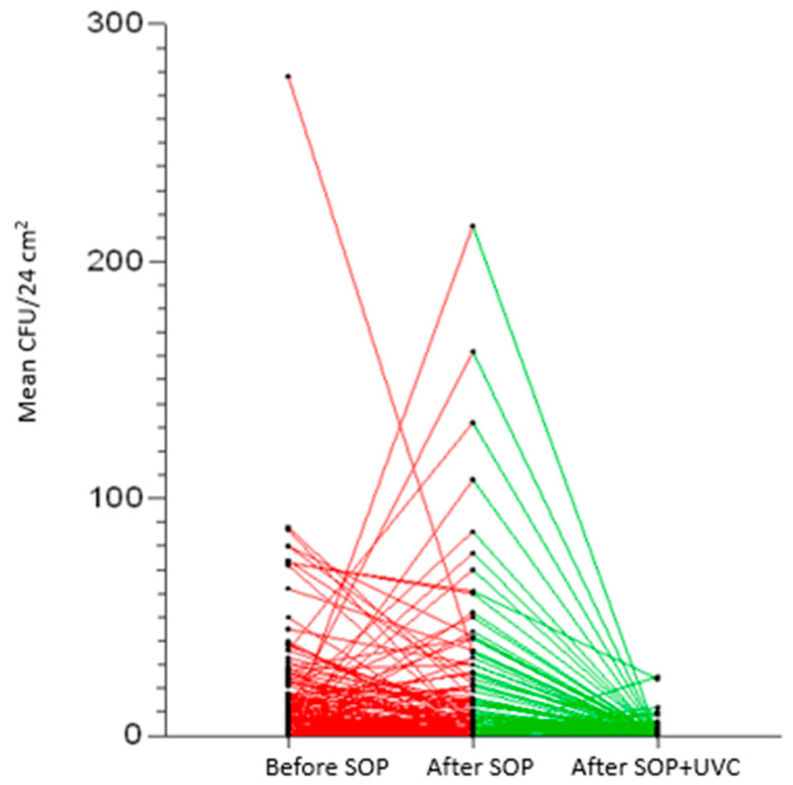
Total viable counts (TVCs) reduction chart after each treatment. In red the variation of the TVCs between before and after SOP; in green the variation of the TVCs between after SOP and after SOP+UV-C.

**Figure 2 ijerph-20-04284-f002:**
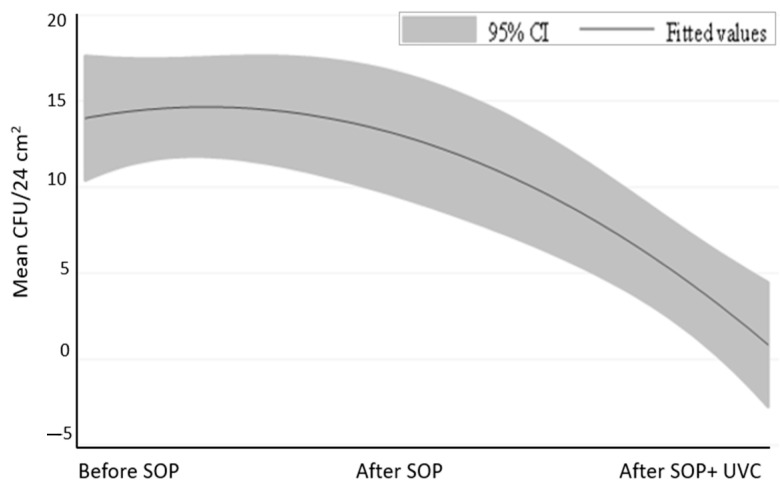
Prediction plot of quadratic regression model with [95% CI]. TVC (variable employee) and type of treatment (independent variable).

## Data Availability

The data were collected and processed directly by the authors, so we have no link to suggest.

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
