# Peer review of "Implementation of an Environmental Cleaning Protocol in Hospital Critical Areas Using a UV-C Disinfection Robot"

_ijerph, 2023, doi:10.3390/ijerph20054284_

Round 1

Reviewer 1 Report

The manuscript describes an interventional study that considers the effectiveness of UV-C radiation for surface disinfection.

As a general point, there is little that is new in this study, given that UV-C has been used for surface disinfection for nearly 100 years. The authors may wish to bring out any key points that are different to other studies. An important factor is the relative importance of transmission routes - whether airborne or via touched surfaces. I suggest a review of the cited references, some of which refer to air disinfection and not surface disinfection.

Of course, the results are important, but more relevant as a technical note than as an original article.

There are a small number of typographical errors:

Line 37, "originates" should be "originate"

Line 58, "use" should be "used"

Line 64, suggest rewording the start of this line to "the COVID-19 pandemic UV disinfection has been applied to several ..."

Line 65, full stop missing after "[9,10]"

Line 89, "declare" should be "declares"

Line 90, if the wavelength range emitted by the lamp is really 100-280 nm then it is quite likely that ozone gas will be produced. It would have been more informative if the emission spectrum had been measured. 

Line 94, insert "operating" after "standard"

Line 98, "chorin" should be "chorine"

Line 112, "sites" should be "site"

Line 190, "as well as" is repeated - delete one occurence

Line 199, "same" should be "some"

Lines 201/2 "applicate" should be "applied"

Line 204, "pathogen" should be "organism"

Line 222, should the "o" before "patient" be "or"?

Line 236, there is something missing. Perhaps "this" should be inserted before "is assigned"

Author Response

Dear Reviewer, 

thank you very much for the attention paid to our manuscript. Below are the answers to your questions and the changes are marked in red in the text.

With kind regards, 

Beatrice Casini

Review 1

R: Line 37, "originates" should be "originate"

A: Thank you for this suggestion. The term "originates" has been replace with "originate”.

R: Line 58, "use" should be "used"

A: Thank you for this suggestion. The term "use" has been replaced with "used."

R: Line 64, suggest rewording the start of this line to "the COVID-19 pandemic UV disinfection has been applied to several ..."

A: Thank you for this suggestion. The sentence has been modified as suggest by the reviewer.

R: Line 65, full stop missing after "[9,10]"

 A: Thank you for this suggestion. The full stop has been added.

R: Line 89, "declare" should be "declares"

A: Thank you for this suggestion. The term "declare" has been replaced with "declares."

R: Line 90, if the wavelength range emitted by the lamp is really 100-280 nm then it is quite likely that ozone gas will be produced. It would have been more informative if the emission spectrum had been measured.

A: Thank you for pointing it out. The ozone emission had not been recorded because the robot is equipped with amalgam lamps with ozone-free patent, declared by manufacturer in the technical sheet   The references of the lamps are reported in the text.

R: Line 94, insert "operating" after "standard"

A: Thank you for this suggestion. The term "operating" was added.

R: Line 98, "chorin" should be "chorine"

A: Thank you for this suggestion. The term "chlorin" has been replaced with "chlorine."

R: Line 112, "sites" should be "site"

A: Thank you for this suggestion. The term "sites" has been replaced with "site."

R: Line 190, "as well as" is repeated - delete one occurrence

A: Thank you for this suggestion. The repetition has been removed.

R: Line 199, "same" should be "some"

A: Thank you for this suggestion. The term "same" has been replaced with "some."

R: Lines 201/2 "applicate" should be "applied"

A: Thank you for this suggestion. The term "applicate" has been replaced with "applied."

R: Line 204, "pathogen" should be "organism"

A: Thank you for this suggestion. The term "pathogen" has been replaced with "microorganism."

R: Line 222, should the "o" before "patient" be "or"?

A: Thank you for this suggestion. The term "o" has been replaced with "or"

R:  Line 236, there is something missing. Perhaps "this" should be inserted before "is assigned.

A: Thank you for this suggestion. The term “this” has been added before “is assigned.”

Reviewer 2 Report

Beatrice Casini, et al. " Implementation of an environmental cleaning protocol in hospital critical areas using a UV-C disinfection robot" This study showed that it was the high effectiveness of UV-C treatment, so there was distinct superiority by UV-C disinfection simultaneously combined the implementation of the standard cleaning and disinfection procedure in the reduction of hygiene failures.

 It seemed that this paper has certain practical reference value and innovation, but the quality of the paper needs to be improved and the writing level needs to be polished.

1.     Please pay attention to the standardized writing of abstracts of journal papers,the following abstract is for reference only:

There is one of the core components to reducing healthcare-associated infections for improving the cleaning and disinfection of high-touch surfaces. In order to evaluate the effectiveness of an enhanced protocol applying the UV-C disinfection for terminal room disinfection between occupying patients, 20 high-touch surfaces in different critical areas were sampled according to ISO 14698-1, immediately pre- and post- standard operating protocol (SOP) and after UV-C disinfection (160 sampling sites in each condition, 480 in total) and the doses emitted at the sites were assessed by dosimeters. It was found that only 17.5% of sampling sites collected after UV-C treatment was positive while 64.3% after SOP treatment. According to the national hygienic standards for health-care setting, only 1.2% of sampling sites collected resulted non-compliant after UV-C disinfection while 9.3% after SOP disinfection. There was 1.6% of the least compliant with the standard limit (≥15 Colony-forming unit/24 cm2) after UV-C treatment while 12% after SOP treatment in the operation theatres. the results showed that it was the high effectiveness of UV-C treatment, so there was distinct superiority by UV-C disinfection simultaneously combined the implementation of the standard cleaning and disinfection procedure in the reduction of hygiene failures.

2.     Report of results and statistical analyses needs to be accurate and rigorous:

Are these two sentences contradictory? One sentence is “: ……with the standard limit (≥15 Colony-forming unit/24 cm2 ) in P1(22), the other is “ ……that recommended for ICUs a limit ≤50 CFU/24 cm2 and ≤15 CFU/24 cmfor operating theatres in P3(109).

 P2(73-74): Personally, I feel that this sentence “when applied in addition to standard cleaning and disinfection protocols in critical areas with a high risk of transmission of healthcare infections.” is ambiguous. It really means something else, please explain or verify it.

 What is the abbreviation of SOP? SOP is abbreviation of the standard operating protocol in the abstract. standard cleaning and disinfection protocols? cleaning and disinfection standard protocol (SOP)?

 P3(100): Through this sentence of “After SOP, the exposure time of the UV-C Robot was 5 minutes for each cycle”, I'm a little confused, what is the experimental relationship between SOP and UV-C in your two comparative experiments? I want to confirm whether SOP is used alone, UV-C is used alone or UV-C used after SOP? If they are connected in series, they cannot be directly used “after SOP” and “after UV-C” for comparison and analysis

 P5-6(169-207): Is it more appropriate to put this part in the preface than here (in line 169-207)? The content of the discussion section should focus on the experimental results.

 3. Language problem:

It is recommended that the authors seek assistance in proof-reading the manuscript before submission.
4. Tables and Figures:
P4(160-161): Is there reasonable in the position of the marks in Figure 1 and Figure 2? In addition, are you sure that the ordinates in Figure 1 and Figure 2 are really the same?

 P4(164): I completely do not understand the information to be expressed in Figure 1, nor can I match the content discussed in the results. Whether the results are completely covered in Figure 1 because you were discussed in the OT, the ICU isolation room and the patient rooms, respectively. Personally, do you think that the description of total viable count (TVC) reduction can be clearly expressed in a table or more Figures in different experimental condition?

P5-6(169-207): Is it more appropriate to put this part in the preface than here (in line 169-207)? The content of the discussion section should focus on the experimental results.

Author Response

Dear Reviewer,

thank you very much for your attention to our article. Below are the answers to your questions and the changes are marked in red in the text.

Best regards

Beatrice Casini

Reviewer 2:

R: Please pay attention to the standardized writing of abstracts of journal papers, the following abstract is for reference only

A: Thank you for this suggestion. The authors modified the abstract according to journal guidelines making it a single paragraph having deleted the headings.

R: Are these two sentences contradictory? One sentence is “: ……with the standard limit (≥15 Colony-forming unit/24 cm2 ) in P1(22), the other is “ ……that recommended for ICUs a limit ≤50 CFU/24 cm2 and ≤15 CFU/24 cm2 for operating theatres in P3(109).

A: Thank you for this suggestion. The symbol has been corrected in the abstract, accordingly with the guidelines.

R: P2(73-74): Personally, I feel that this sentence “when applied in addition to standard cleaning and disinfection protocols in critical areas with a high risk of transmission of healthcare infections.” is ambiguous. It really means something else, please explain or verify it.

A: Thank you for the comment. This sentence describes the aim of the study indicating the level of risk of the selected hospital settings. UV-C treatment is particularly efficient in critical settings, where the risk of transmission of healthcare infections is especially high, but it can also be applied in other settings. In any case, UV-C treatments should be considered as an implementation of the standard protocol and not as a replacement.

R: What is the abbreviation of SOP? SOP is abbreviation of the standard operating protocol in the abstract. standard cleaning and disinfection protocols? cleaning and disinfection standard protocol (SOP)?

A: Thank you for pointing it out. We added throughout the text the definition of standard operating protocol: “cleaning and disinfection standard operating protocol, (SOP)”.

R: P3(100): Through this sentence of “After SOP, the exposure time of the UV-C Robot was 5 minutes for each cycle”, I'm a little confused, what is the experimental relationship between SOP and UV-C in your two comparative experiments? I want to confirm whether SOP is used alone, UV-C is used alone or UV-C used after SOP? If they are connected in series, they cannot be directly used “after SOP” and “after UV-C” for comparison and analysis.

A: Thank you for this suggestion. The two treatments are performed in series and the sampling is carried out immediately after SOP alone, and after SOP + UV-C. The use of UV-C alone was not considered in our study.  We added the sentence throughout the text, following your suggestion.

R: P5-6(169-207): Is it more appropriate to put this part in the preface than here (in line 169- 207)? The content of the discussion section should focus on the experimental results.

A: Thank you for this suggestion. This part has been moved from discussion section to introduction, as suggested.

R: P4(160-161): Is there reasonable in the position of the marks in Figure 1 and Figure 2? In addition, are you sure that the ordinates in Figure 1 and Figure 2 are really the same?

A: Thank you for this suggestion. The ordinate represented in Figure 1 and Figure 2 are the same. We have modified the caption, making it uniform in the two figures.

R: P4(164): I completely do not understand the information to be expressed in Figure 1, nor can I match the content discussed in the results. Whether the results are completely covered in Figure 1 because you were discussed in the OT, the ICU isolation room and the patient rooms, respectively. Personally, do you think that the description of total viable count (TVC) reduction can be clearly expressed in a table or more Figures in different experimental condition?

A Thank you for your comment. The choice of represent the TVC reduction of all settings data in Figure 1 instead of separately in the different settings, as reported in the text, was due to the non-equal sample size. In fact, the number of sampled sites in ICU and patient rooms was much smaller than in the OTs. Only the global view of TVC reduction in all sites allows to appreciate the effectiveness of UV-C disinfection.
